# Candida Carriers among Individuals with Tongue Piercing—A Real-Time PCR Study

**DOI:** 10.3390/antibiotics11060742

**Published:** 2022-05-31

**Authors:** Georgi Tomov, Nikola Stamenov, Deyan Neychev, Kiril Atliev

**Affiliations:** 1Department of Periodontology and Oral Mucosa Diseases, Faculty of Dental Medicine, Medical University of Plovdiv, 15-A “Vasil Aprilov” Blvd, 4002 Plovdiv, Bulgaria; nikola.stamenov@mu-plovdiv.bg; 2Department of Oral Surgery, Faculty of Dental Medicine, Medical University of Plovdiv, 15-A “Vasil Aprilov” Blvd, 4002 Plovdiv, Bulgaria; deyan.neychev@mu-plovdiv.bg; 3Department of Urology and General Medicine, Faculty of Medicine, Medical University of Plovdiv, 15-A “Vasil Aprilov” Blvd, 4002 Plovdiv, Bulgaria; kiril.atliev@mu-plovdiv.bg

**Keywords:** tongue piercing, *Candida* spp., RT-PCR

## Abstract

Among the local factors for oral candidiasis, the piercing of the tongue is recognized by some authors as a risk factor for the colonization of *Candida albicans*. There are few case reports in which *Candida* spp. colonization and infection are associated with tongue piercing but only one microbiological study supports this hypothesis in general. The aim of this study was to examine this possible association between the presence of both tongue piercing and *Candida* spp. in healthy individuals. Positive results for tongue colonization with *Candida* spp. were found in four (12.9%) of the tongue-pierced subjects and in three (9.67%) subjects of the control group (*p* = 0.550). All samples were identified as *Candida albicans*. The univariate and logistic regression analyses of possible risk factors for tongue colonization revealed that gender (*p* = 0.024), smoking more than 10 cigarettes per day (*p* = 0.021), and improper hygiene (*p* = 0.028) were statistically significant influencing factors in the multivariate analysis. The results suggest that the piercing of the tongue is not a risk factor for colonization of *Candida* spp.

## 1. Introduction

*Candida* spp. are among the first colonizers in the oral cavity and their presence as saprophytic organisms is not considered a health problem in immunocompetent individuals. A better understanding of the nuances of human immune systems has revealed that oral mucosa immunity delivers a unique response to fungal pathogens. Oral fungal infection does not depend solely on the fungus and the host, however, and attention has now focused on interactions with other members of the oral microbiome. It is evident that there is inter-kingdom signaling that affects microbial pathogenicity [1]. When the oral eubiosis is altered, the dynamic and polymicrobial oral microbiome could be a direct precursor of different diseases including oral candidiasis. As a typical opportunistic infection, oral candidiasis occurs when systemic factors that interfere with temporary or constant immunodeficiency (HIV, oncological disease, autoimmune disease, etc.) or local factors—such as poor hygiene, denture wearing, xerostomia, and topical applications of corticosteroid sprays—benefit the overgrowing of *Candida* spp. in the oral cavity [2].

Adherence is the first step in colonization [3] and the adherence of *C. albicans* cells to a variety of substrates, including buccal cells [4] and dental acrylic [5], has been investigated. It seems that different objects in the oral cavity—such as dental appliances, dentures, and oral piercing—could harbor *Candida* spp., especially when these objects are retentive (acrylic base) and are isolated from the direct rinsing action of salivary flow [6] or when hygiene is neglected [7]. Among the local factors, the piercing of the tongue is recognized by some authors as a risk factor for the colonization of *Candida albicans* [8]. There are few case reports in which *Candida* spp. colonization and infection are associated with tongue piercing [9,10], but only one microbiological study supports this hypothesis in general [8].

The disputable point is the potential role of oral piercing as an ecological niche and factor modulating virulence in *Candida* species, potentially turning it into a pathobiont under conditions of ecological imbalance. This potential depends on many factors, but the presence of *Candida* species (and their increased number) in the piercing tunnel is considered crucial for further development of oral candidiasis.

The aim of this study was to examine this possible association between the presence of both tongue piercing and *Candida* spp. in healthy individuals. Our null hypothesis is that there will be an elevation in the prevalence of *Candida* spp. colonization as tongue-piercing sites can serve as a retention and colonization ecological niche.

## 2. Results

The inclusion criteria were fulfilled by 62 patients who were enrolled in the study. Among them, 31 constituted the study group of patients with tongue piercings. The other 31 are included in the control group (patients without tongue piercings).

The mean age (±s.d.) of the study and comparison groups was 23.83 (±5.06) and 23.16 (±1.7) years (range 18–37 years), respectively, and mean time (±s.d.) from piercing was 83.16 (±3.47) months. Women with piercings in the study accounted for 76.2% (26 cases) and men for 23.8% (5 cases). Gender and age distribution in the control group were identical. Positive results for tongue colonization with *Candida* spp. were found in four (12.9%) of the tongue-pierced subjects and in three (9.67%) subjects of the comparison group (*p* = 0.550). All samples were identified as *Candida albicans*.

The univariate and logistic regression analyses of possible risk factors for tongue colonization are presented in Table 1. Gender (*p* = 0.024), smoking more than 10 cigarettes per day (*p* = 0.021), and improper hygiene (*p* = 0.028) were statistically significant influencing factors in the multivariate analysis.

## 3. Discussion

The oral cavity is a unique ecological niche for microbial colonization. It provides a variety of surfaces for colonization ranging from the hard non-shedding surfaces of teeth to desquamating keratinized and non-keratinized epithelia. The surfaces in the mouth are kept warm and moist by the constant flow of saliva across them. It is not surprising, therefore, that the human oral cavity supports a complex and dynamic microbiota [11]. In general, this microbiota in healthy individuals is non-pathogenic and may indeed prevent colonization by overtly pathogenic microorganisms. However, some individuals have dental appliances which introduce acrylic, ceramic, and metal alloy surfaces that are also colonized (Figure 1). Oral piercings are a specific foreign body object in the oral cavity.

Oral piercing is a practice that is gaining popularity as a sign of individuality or membership in specific social groups [12]. Its prevalence is changing constantly due to trends’ fluctuations. In the Israeli youth population, for example, the prevalence of oral piercing is reported as being between 3.4% and 20.3% [13]. However, the Candida-related complications of tongue piercing are considered rare with only two reported clinical cases [9,10]. The first case reported the appearance of symptoms following a tongue piercing insertion in a young woman. The case is interpreted by the authors as an acute infection [9]. *C. albicans*, *C. tropicalis*, *C. glabrata*, and *C. krusei* were identified from the sample. The treatment included piercing removal and combined antifungal and antibiotic therapy. The second clinical case addresses a 20-year-old healthy, white woman in a stable exclusively lesbian relationship. She was consulted together with her female partner for recurring vaginal infections of fungal origin. The patient also reported problems in the oral cavity related to her tongue piercing. The oral examination revealed that the piercing site is red and covered with a whitish coating. The burning sensation and the mouth soreness were evident for 10 days. Mycological testing revealed the presence of *C. dubliniensis* in both the patient and her partner. The patient was treated topically by vaginal lavage with boric acid and with oral nystatin suspension for 2 weeks. Her symptoms improved significantly after 2 weeks, and her test was negative after 2 months [10]. The presented clinical cases should not be classified as simple tongue piercing carriers. The acute infection in the first case and the presence of related comorbidity in the second case are rather a background for the development of candida infection.

Patients with tongue piercings in the present study were immunocompetent young individuals without accompanying systemic or local risk factors. The prevalence of *Candida* spp. colonization in the study group was not found to be statistically different from the control group of patients without tongue piercings. In our study, we have chosen a real-time PCR for determining the *Candida* species. Real-time PCR detects the accumulation of amplicon during the reaction. The data is then measured at the exponential phase of the PCR reaction. Traditional PCR methods use agarose gels or other post PCR detection methods, which are not as precise. Real-time PCR makes quantitation of DNA and RNA easier and more precise than other methods. In our study, all samples were identified as colonies of *Candida albicans*, which is in agreement with a previous study conducted by Zadik Y. et al. [8]. In their study, however, the authors used Chromagar media for this purpose and this is a limitation declared by the authors themselves [8]. According to their interpretation, the precise differentiation between *Candida albicans* and *Candida dubliniensis* is not achievable by using this methodology [8]. Regarding the effect of smoking on *Candida* spp. Colonization, our findings are in agreement with previous studies [8,14].

The low percentage of positive *Candida* spp. carriers in our study (12.9%) in comparison with other reports is a conflicting point for discussion. The very obvious explanation for this controversy is the health status of the reported groups. Generally, the reports are focused on risky groups such as newly born [15] and very old persons [16] as well as non-immunocompetent individuals, patients with poorly controlled diabetes or patients treated with antibiotics, corticosteroids, immunosuppressors, or xeroinductors [10]. Furthermore, the mean time (±s.d.) from piercing was in our study is 83,16 (±3.47) and 26.0 (±19.8) months respectively in the study of Zadik Y. et al. [8]. However, the number of the subjects in our study is very low, and we consider this as a limitation for the data interpretation.

Our results raise a few questions. It is proven that the biofilm-forming capacity and virulence of *Candida* spp. have intrinsically heterogeneous features [17]. Nevertheless, the conditions of the ecological environment or niche may impact or condition its potential virulence, probably via epigenetic mechanisms [18,19]. In this context, the presence of a tongue piercing is a potential factor whose impact remains unclear. Some studies suggest that *Candida* spp. sensu stricto could work as an accelerator of periimplantitis [20,21]. These results are intriguing because of the fact that both oral piercing and dental implants are foreign objects in the oral cavity. The material is also is also considered crucial for the colonization of *Candida* spp. According to Devcic M.K. et al., subjects who have PMMA-based dentures more frequently exhibit Candida colonization, with *C. albicans* being the predominant species [16]. Subjects with metal framework-based dentures were less prone to Candida colonization and had better values of salivary flow rate [16]. However, this hypothesis needs to be proven in clinical studies, particularly longitudinal and prospective ones.

Additionally, a literature review of machine-learning-based diagnosis and prognosis in clinical dentistry found reports of the use of machine learning algorithms in periodontics and oral medicine [22]. Machine learning has been been used to integrate microbiome data with immune profiling to stratify peri-implantitis patients according to clinical outcomes [23]. There are exciting future prospects of incorporating a wider range of datasets in AI approaches to improve the diagnosis of, and predict risk from, oral fungal infections in patients with oral piercings.

## 4. Materials and Methods

### 4.1. Study Population

The participants in this study were healthy young adults who were outpatients of the Faculty of Dental Medicine in Plovdiv, Bulgaria. Inclusion criteria included the presence of tongue oral piercing (metal, acrylic, or combined). The exclusive criteria were focused on the lack of symptoms from the tongue piercing such as inflammation or functional disturbances. Additionally, the use of removable oral appliances was considered an exclusive criterion. Special attention was paid to the administration of different topical and systemic drugs such as antibiotics, corticosteroids, immunosuppressants, and xeroinductors. Patients with systemic diseases—including diabetes, HIV, chronic infections, autoimmune diseases, and active cancer therapy—were excluded from the study. The enrolled patients who covered the requirements were informed in detail about the goal of the study and individual informed consent was obtained.

The control group comprises healthy patients of the same age/gender. Data including age, gender, smoking, oral hygiene practice, and time from piercing procedure were collected from each participant.

### 4.2. Laboratory Methods

Samples were collected from the piercing tunnel of each subject in the study group and from an estimated anterior third of the dorsal tongue in the control group using a sterile paper point (CAT plus, MIP Pharma, Germany) (Figure 2).

Samples were sent to a certified laboratory in Germany for qualitative and quantitative RT-PCR analysis (Figure 3).

### 4.3. Statistical Analysis

The prevalence of *Candida* spp. in the soft tissue tunnel formed by tongue piercing was evaluated and compared with the control group by using the chi-square test. Additionally, a *Candida* spp. colonization logistic regression model was performed in relation to patients’ gender, smoking, and oral hygiene habits as explanatory variables.

Data were analyzed by SPSS 15.0 (SPSS, Inc., Chicago, IL, USA) statistical software. Two-sided *p* < 0.05 was considered statistically significant.

The study design was approved by the Ethics Committee of the Medical University of Plovdiv as part of a PhD project entitled “Piercing and oral health” (Protocol No. 5/29. 10. 2015).

## 5. Conclusions

The analysis of host–microbe interactions has advanced markedly in recent decades, but the key question concerning the ability to predict oral fungal infections in individuals at risk still remains. The role of the oral piercing as an ecological niche and another key factor modulating virulence of *Candida* species, potentially turning them into a pathobiont under conditions of ecological imbalance, is disputable. Previous studies hypothesized the presence of *Candida* spp. in the oral piercing tunnel as a potential risk factor for the development of oral candidiasis. With all limits of this study, our results do not support this hypothesis. The low percentage of positive *Candida* spp. carriers (12.9%) and the lack of clinical signs for candida-induced infection revealed that the systemic factors play a more important role in this process than the local retentive factors such as oral piercing. However, the obtained data could be extended in future and incorporated into machine learning algorithms. The integration of microbiological data with immune profiling and other systemic factors will improve the diagnosis of, and predict risk from, oral fungal infections in patients with oral piercing.

## Figures and Tables

**Figure 1 antibiotics-11-00742-f001:**
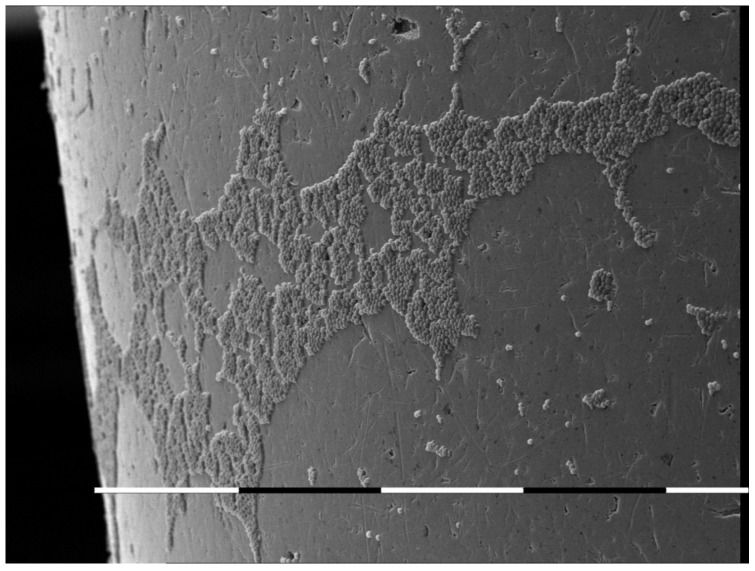
In vitro colonization of metal oral piercing with *Candida albicans*. SEM picture [6].

**Figure 2 antibiotics-11-00742-f002:**
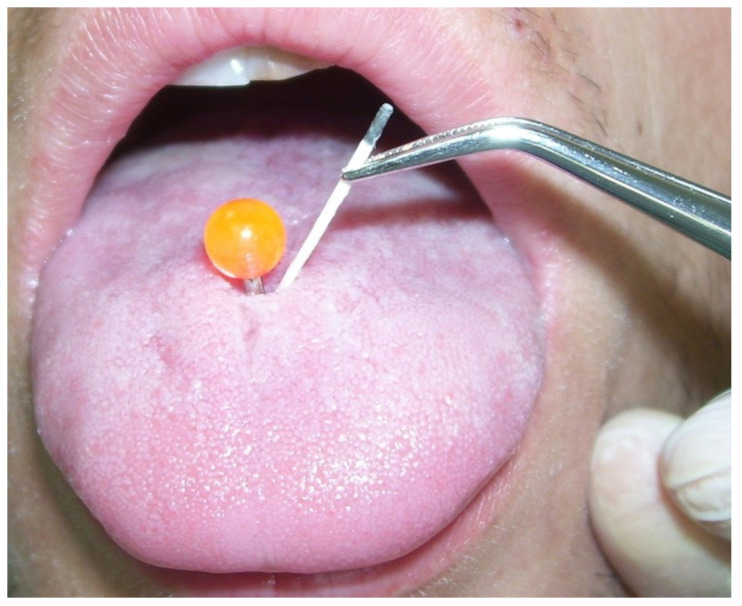
A sampling of the tongue piercing tunnel.

**Figure 3 antibiotics-11-00742-f003:**
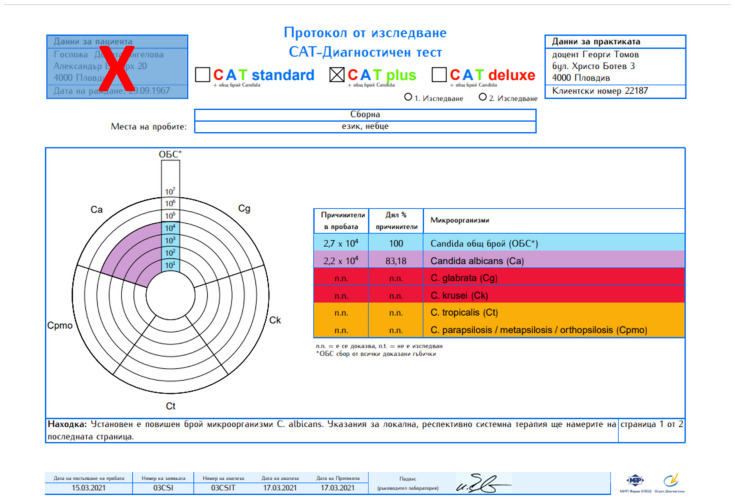
An anonymized result from CAT plus test revealing presence of *C. albicans*.

**Table 1 antibiotics-11-00742-t001:** Univariate and logistic regression analysis of possible risk factors for Candida colonization in the study population.

Factor	*n*	Colonizationn%	Univariate Analysis			Logistic Regression Analysis		
			OR	95% CI	*p*-Value	OR	95% CI	*p*-Value
Tongue piercing								
No	31	3 (9.67%)	1.000	0.441–2.743		1.000	0.519–3.423	
Yes	31	4 (12.9%)	1.200		1.000	1.333		0.550
Gender								
M	10	3 (30%)	1.000	0.949–6.749		1.300	1.172–9.472	
F	52	4 (7.69%)	2.578		0.069	3.332		0.024 *
Smoking				0.434–2.758 0.533–2.851			0.522–3.421 1.169–9.475	
0	26	1 (3.84%)	1.200		0.068	1.000		0.550
1–10	20	2 (10%)	1.100			1.332		
>10	16	4 (25%)	2.100		0.683	3.334		0.021 *
Tongue brushing								
Yes	43	2 (4.65%)	1.100	0.952–6.754		1.333	1.175–9.469	
No	19	5 (26.31%)	2.536		0.069	3.331		0.028 *

OR (odds ratio); CI (confidence interval). * Statistically significant.

## Data Availability

According to the Bulgarian law, data on patients’ health are considered “sensitive”. The ethics committee asked the researcher responsible (G. Tomov) to sign a document where he is committed to protecting the data and to communicate with the ethics committee regarding any requests on the matter.

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
