# Peer review of "Candida Carriers among Individuals with Tongue Piercing—A Real-Time PCR Study"

_antibiotics, 2022, doi:10.3390/antibiotics11060742_

Round 1

Reviewer 1 Report

The manuscript must be reviewed because it does not bring anything innovative. The introduction, discussion and conclusions are rather flat. The authors should make a large change in the introduction, discussion and conclusions to get the manuscript fit for publication. The discussion should be expanded to included a weaknesses paragraph. Data description and presentation is very confusing. Furthermore, the research results are not scientifically relevant. The statistical analysis is poor. There are grammatical errors throughout the manuscript.

Author Response

Dear Madam/Sir,

We appreciate your efforts and precious time in reviewing our paper and providing valuable comments. All authors have carefully considered the remarks and tried their best to address every one of them.

Please see the  point-by-point responses (highlighted in red) and the revised paper.

Point 1: The manuscript must be reviewed because it does not bring anything innovative. The introduction, discussion and conclusions are rather flat. The authors should make a large change in the introduction, discussion and conclusions to get the manuscript fit for publication.

Response 1: We agree with the reviewer’s comments regarding some weaknesses and insufficiencies of the introduction, discussion and conclusion. These parts were rewritten in order to present the main idea, namely the hypothesized role of oral piercing as a risk factor for the development of oral candidiasis. Additionally, 12 new publications were added to the list of references to enrich the discussion with new information and interpretations.

Point 2: The discussion should be expanded to included a weaknesses paragraph. Data description and presentation is very confusing. Furthermore, the research results are not scientifically relevant. The statistical analysis is poor. There are grammatical errors throughout the manuscript.

Response 2: The reviewer’s comments were taken into consideration and two new authors were invited to join the research team and to contribute with their expertise in nosocomial infections and biostatistics. Their valuable analysis and interpretation of the results in the discussion improved the quality of the text. Weaknesses paragraph was added too. Grammar corrections were done accordingly.

Reviewer 2 Report

The manuscript entitled "Candida Carriers among Individuals with Tongue Piercing – A 2 Real-Time Pcr Study" aim to evaluate the 
possible association between the presence of both tongue piercing and Candida spp. in healthy individuals. The topic is very interesting however the manuscript lack of the appropiate introduction. The english is very poor. The methodology appear appropiate. The discussion section is too short. The author did not explain the importance of the study. I suggest to reconsider this manuscript after major revision. 

Author Response

Dear Madam/Sir,

We appreciate your efforts and precious time in reviewing our paper and providing valuable comments. All authors have carefully considered the remarks and tried their best to address every one of them.

Please see the  point-by-point responses (highlighted in red) and the revised paper.

Point 1: The manuscript entitled "Candida Carriers among Individuals with Tongue Piercing – A 2 Real-Time Pcr Study" aim to evaluate the possible association between the presence of both tongue piercing and Candida spp. in healthy individuals. The topic is very interesting however the manuscript lack of the appropiate introduction. The english is very poor. The methodology appear appropiate. The discussion section is too short. The author did not explain the importance of the study. I suggest to reconsider this manuscript after major revision.

Response 1: We agree with the reviewer’s comments regarding some weaknesses and insufficiencies of the introduction and discussion sections. These parts were rewritten in order to present the main idea, namely the hypothesized role of oral piercing as a risk factor for the development of oral candidiasis. Additionally, 12 new publications were added to the list of references to enrich the discussion with new information and interpretations. Two new authors were invited to join the research team and to contribute with their expertise in the field of nosocomial infections and biostatistics. Their valuable analysis and interpretation of the results in the discussion improved the quality of the text. Grammar corrections were done accordingly.

Reviewer 3 Report

This work studies the possible Candida carriage in tongue pierced individuals. The work analyses the possible link between the presence of both tongue piercing and Candida spp. in healthy individuals. The results indicate tongue colonization with Candida albicans 12.9% of the subjects. Gender, smoking more than 10 cigarettes per day, and improper hygiene were statistically significant rick factors. Yet, it is concluded that tongue piercing is not a risk factor for colonization of Candida spp.

In general, the work seems to be well performed, and it is interesting, but there are several critical points that need to be carefully adjusted/clarified. Critically, Candida ID needs to be much well explained and justified (Why RT PCR and not PCR?). Also, discussion should be deeper. It should be, for example, very interesting to compare Candida carriage in piercings with studies related to denture (or other mouth prosthetic device carriage).

In detail:

  • Candida needs to be written in italic. Correct this in the entire MS, since this is not OK in all “Candida” mentions;
  • References used to support the MS are old and should be updated. There are several recent and key works that should be used replacing older references, such as: DOI: 2217/fmb-2020-0113; doi: 10.3390/jof7020079;
  • Section 6 (Patents) needs to be cut;
  • Reference for the study (ethical issues) and date needs to be included (is mandatory and the lack of inclusion is a reason for rejection);
  • The number of subjects is very low and needs to be indicated as a limitation of this study.

Introduction:

  • This section is very little and should be expanded a bit;

M&M:

  • Headlines with subsections should be used (e.g. 4.1; 4.2...);
  • Figure 2 is too small and has poor quality. Please adjust;
  • As previously referred: why RT PCR (gene expression quantification) and not PCR (gene identification)? It is quite strange. How was the RT PCR performed? This needs to be well clarified.

Author Response

Dear Madam/Sir,

We appreciate your efforts and precious time in reviewing our paper and providing valuable comments. All authors have carefully considered the remarks and tried their best to address every one of them.

Please see the point-by-point responses (highlighted in red) and the revised paper.

Point 1: In general, the work seems to be well performed, and it is interesting, but there are several critical points that need to be carefully adjusted/clarified. Critically, Candida ID needs to be much well explained and justified (Why RT PCR and not PCR?).

Response 1: The decision to choose RT PCR instead of PCR was augmented by the differences between the two methods. Real-Time PCR detects the accumulation of amplicon during the reaction. The data is then measured at the exponential phase of the PCR reaction. Traditional PCR methods use agarose gels or other post PCR detection methods, which are not as precise. Real-Time PCR makes quantitation of DNA and RNA easier and more precise than other methods. This test is performed in a certified laboratory in Germany (MIP Pharma GmbH, Department of Diagnostics, Kirkeler Straße 41, D-66440 Blieskastel, Germany) and is an established standard for our Department because we receive both qualitative and quantitative results together with recommendations for antifungal treatment.

Point 2: Also, discussion should be deeper. It should be, for example, very interesting to compare Candida carriage in piercings with studies related to denture (or other mouth prosthetic device carriage).

Response 2: We completely agree with the reviewer’s comments regarding some weaknesses and insufficiencies of the discussion section. For this reason, we enriched the text with information from newly published research by Devcic, M.; et al. “Oral Candidal Colonization in Patients with Different Prosthetic Appliances” (J. Fungi 2021, 7, 662. https://doi.org/10.3390/jof7080662) in which the Candida colonization in patients with different oral appliances is presented in details and was an excellent base for comparison.   

Point 3: In detail:

Candida needs to be written in italic. Correct this in the entire MS, since this is not OK in all “Candida” mentions;

Done

References used to support the MS are old and should be updated. There are several recent and key works that should be used replacing older references, such as: DOI: 2217/fmb-2020-0113; doi: 10.3390/jof7020079;

Twelve new publications were added to the list of references to enrich the discussion with new information and interpretations. Few old publications were replaced with new relevant ones.

Section 6 (Patents) needs to be cut;

Done

Reference for the study (ethical issues) and date needs to be included (is mandatory and the lack of inclusion is a reason for rejection);

The study design was approved by the Ethics Committee of the Medical University of Plovdiv as part of a PhD project entitled “Piercing and oral health” (Protocol No 5/ 29. 10. 2015). According to the national Personal Data Protection Code, data on patients’ health are considered “sensitive”. The Ethics committee asked the main researcher to sign a document where he is committed to protecting the data and to relate to the Ethics committee for any request on the matter.

The number of subjects is very low and needs to be indicated as a limitation of this study.

Weaknesses paragraph was added where the limitations of the study are clearly explained.

Introduction:

This section is very little and should be expanded a bit;

We agree with the reviewer’s comments regarding some weaknesses and insufficiencies of the introduction section. This part was rewritten in order to present the main idea, namely the hypothesized role of oral piercing as a risk factor for the development of oral candidiasis.

M&M:

Headlines with subsections should be used (e.g. 4.1; 4.2...);

Done

Figure 2 is too small and has poor quality. Please adjust;

Figures with better quality are provided.

As previously referred: why RT PCR (gene expression quantification) and not PCR (gene identification)? It is quite strange. How was the RT PCR performed? This needs to be well clarified.

See: Response 1

Round 2

Reviewer 1 Report

The authors made all required changes in Round 1 of review. The manuscript is eligible for publication.

Reviewer 2 Report

The author improve the manuscript as suggested it could be suitable for publication 

Reviewer 3 Report

Dear authors,

Thank you for the clarifications.

Just a final note: figures are not in order (fig 2 appears before fig 1) and fig 2 should be sharper. Please pay attention to the italics.

Anyway, the work was improved.

Good luck.